# Min-Max Propagation

**Christopher Srinivasa**
University of Toronto
Borealis AI
christopher.srinivasa
@gmail.com

**Inmar Givoni**
University of
Toronto
inmar.givoni
@gmail.com

**Siamak Ravanbakhsh**
University of
British
Columbia
siamakx@cs.ubc.ca

**Brendan J. Frey**
University of Toronto
Vector Institute
Deep Genomics
frey@psi.toronto.edu

## Abstract

We study the application of min-max propagation, a variation of belief propagation, for approximate min-max inference in factor graphs. We show that for "any" high-order function that can be minimized in $\mathcal{O}(\omega)$, the min-max message update can be obtained using an efficient $\mathcal{O}(K(\omega + \log(K))$ procedure, where $K$ is the number of variables. We demonstrate how this generic procedure, in combination with efficient updates for a family of high-order constraints, enables the application of min-max propagation to efficiently approximate the NP-hard problem of makespan minimization, which seeks to distribute a set of tasks on machines, such that the worst case load is minimized.

## 1 Introduction

Min-max is a common optimization problem that involves minimizing a function with respect to some variables $X$ and maximizing it with respect to others $Z$: $\min_X \max_Z f(X, Z)$. For example, $f(X, Z)$ may be the cost or loss incurred by a system $X$ under different operating conditions $Z$, in which case the goal is to select the system whose worst-case cost is lowest. In Section 2, we show that factor graphs present a desirable framework for solving min-max problems and in Section 3 we review min-max propagation, a min-max based belief propagation algorithm.

Sum-product and min-sum inference using message passing has repeatedly produced groundbreaking results in various fields, from low-density parity-check codes in communication theory (Kschischang et al., 2001), to satisfiability in combinatorial optimization and latent-factor analysis in machine learning.

An important question is *whether "min-max" propagation can also yield good approximate solutions when dealing with NP-hard problems?* In this paper we answer this question in two parts.

I) Our main contribution is the introduction of an efficient min-max message passing procedure for a generic family of high-order factors in Section 4. This enables us to approach new problems through their factor graph formulation. Section 5.2 leverages our solution for high-order factors to efficiently approximate the problem of makespan minimization using min-max propagation. II) To better understand the pros and cons of min-max propagation, Section 5.1 compares it with the alternative approach of reducing min-max inference to a sequence of Constraint Satisfaction Problems (CSPs).

The feasibility of "exact" inference in a min-max semiring using the junction-tree method goes back to (Aji and McEliece, 2000). More recent work of (Vinyals et al., 2013) presents the application of min-max for junction-tree in a particular setting of the makespan problem. In this paper, we investigate the usefulness of min-max propagation in the loopy case and more importantly provide an efficient and generic algorithm to perform message passing with high-order factors.

## 2 Min-Max Optimization on Factor Graphs

We are interested in factorizable min-max problems $\min_X \max_Z f(X, Z)$ – *i.e.* min-max problems that can be efficiently factored into a group of more simple functions. These have the following properties:

1. The cardinality of either $X$ or $Z$ (say $Z$) is linear in available computing resources (*e.g.* $Z$ is an indexing variable $a$ which is linear in the number of indices)
2. The other variable can be decomposed, so that $X = (x_1, \ldots, x_N)$
3. Given $Z$, the function $f()$ depends on only a subset of the variables in $X$ and/or exhibits a form which is easier to minimize individually than when combined with $f(X, Z)$

Using $a \in \mathcal{F} = \{1, \ldots, F\}$ to index the values of $Z$ and $X_{\partial a}$ to denote the subset of variables that $f()$ depends on when $Z = a$, the min-max problem can be formulated as,

$$\min_X \max_a f_a(X_{\partial a}). \tag{1}$$

In the following we use $i, j \in \mathcal{N} = \{1, \ldots, N\}$ to denote variable indices and $a, b \in \{1, \ldots, F\}$ for factor indices. A Factor Graph (FG) is a bipartite graphical representation of the above factorization properties. In it, each function (*i.e.* factor $f_a$) is represented by a square node and each variable is represented by a circular node. Each factor node is connected via individual edges to the variables on which it depends. We use $\partial i$ to denote the set of neighbouring factor indices for variable $i$, and similarly we use $\partial a$ to denote the index set of variables connected to factor $a$.

This problem is related to the problems commonly analyzed using FGs (Bishop, 2006): the sum-product problem, $\sum_X \prod_a f_a(X_{\partial a})$, the min-sum problem, $\min_X \sum_a f_a(X_{\partial a})$, and the max-product problem, $\max_X \prod_a f_a(X_{\partial a})$ in which case we would respectively take product, sum, and product rather than the max of the factors in the FG.

When dealing with NP-hard problems, the FG contains one or more loop(s). While NP-hard problems have been represented and (approximately) solved directly using message passing on FGs in the sum-product, min-sum, and max-product cases, to our knowledge this has never been done in the min-max case.

## 3 Min-Max Propagation

An important question is how min-max can be computed on FGs. Consider the sum-product algorithm on FGs which relies on the sum and product operations satisfying the distributive law $a(b + c) = ab + ac$ (Aji and McEliece, 2000).

Min and max operators also satisfy the distributive law: $\min(\max(\alpha, \beta), \max(\alpha, \gamma)) = \max(\alpha, \min(\beta, \gamma))$. Using $(\min, \max, \Re)$ semiring, the belief propagation updates are as follows. Note that these updates are analogous to sum-product belief propagation updates, where sum is replaced by min and product operation is replaced by max.

**Variable-to-Factor Messages.** The message sent from variable $x_i$ to function $f_a$ is

$$\mu_{ia}(x_i) = \max_{b \in \partial i \setminus a} \eta_{bi}(x_i) \tag{2}$$

where $\eta_{bi}(x_i)$ is the message sent from function $f_b$ to variable $x_i$ (as shown in Fig. 1) and $\partial i \setminus a$ is the set of all neighbouring factors of variable $i$, with $a$ removed.

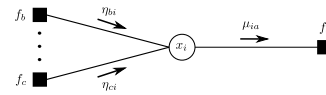

Figure 1: Variable-to-factor message.

**Factor-to-Variable Messages.** The message sent from function $f_a$ to variable $x_i$ is computed using

$$\eta_{ai}(x_i) = \min_{X_{\partial a \setminus i}} \max \left( f_a(X_{\partial a}), \max_{j \in \partial a \setminus i} \mu_{ja}(x_j) \right) \tag{3}$$

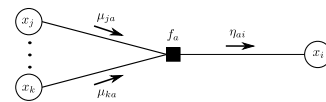

Figure 2: Factor-to-variable message.

**Initialization Using the Identity.** In the sum-product algorithm, messages are usually initialized using knowledge of the identity of

the product operation. For example, if the FG is a tree with some node chosen as a root, messages can be passed from the leaves to the root and back to the leaves. The initial message sent from a variable that is a leaf involves taking the product for an empty set of incoming messages, and therefore the message is initialized to the *identity* of the group $(\Re^+, \times)$, which is $\overset{\times}{1} = 1$.

In this case, we need the identity of the $(\Re, \max)$ semi-group, where $\max(\overset{\max}{1}, x) = x \, \forall x \in \Re$ – that is $\overset{\max}{1} = -\infty$. Examining Eq. (3), we see that the message sent from a function that is a leaf will involve maximizing over the empty set of incoming messages. So, we can initialize the message sent from function $f_a$ to variable $x_i$ using $\eta_{ai}(x_i) = \min_{X_{\partial a \setminus x_i}} f_a(X_{\partial a})$.

**Marginals.** Min-max marginals, which involve "minimizing" over all variables except some $x_i$, can be computed by taking the max of all incoming messages at $x_i$ as in Fig. 3:

$$m(x_i) = \min_{X_{\mathcal{N} \setminus i}} \max_a f_a(X_{\partial a}) = \max_{b \in \partial i} \eta_{bi}(x_i) \qquad (4)$$

The value of $x_i$ that achieves the global solution is given by $\arg \min_{x_i} m(x_i)$.

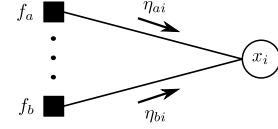

Figure 3: Marginals.

# 4 Efficient Update for High-Order Factors

When passing messages from factors to variables, we are interested in efficiently evaluating Eq. (3). In its original form, this computation is exponential in the number of neighbouring variables $|\partial a|$. Since many interesting problems require high-order factors in their FG formulation, many have investigated efficient min-sum and sum-product message passing through special family of, often sparse, factors (*e.g.* Tarlow et al., 2010; Potetz and Lee, 2008).

For the time being, consider the factors over binary variables $x_i \in \{0, 1\} \forall i \in \partial a$ and further assume that efficient minimization of the factor $f_a$ is possible.

**Assumption 1.** *The function* $f_a : X_{\partial a} \to \Re$ *can be minimized in time* $\mathcal{O}(\omega)$ *with any subset* $\mathcal{B} \subset \partial a$ *of its variables fixed.*

In the following we show how to calculate min-max factor-to-variable messages in $\mathcal{O}(K(\omega + \log(K)))$, where $K = |\partial a| - 1$. In comparison to the limited settings in which high-order factors allow efficient min-sum and sum-product inference, we believe this result to be quite general.[1]

The idea is to break the problem in half, at each iteration. We show that for one of these halves, we can obtain the min-max value using a single evaluation of $f_a$. By reducing the size of the original problem in this way, we only need to choose the final min-max message value from a set of candidates that is at most linear in $|\partial a|$.

**Procedure.** According to Eq. (3), in calculating the factor-to-variable message $\eta_{ai}(x_i)$ for a fixed $x_i = c$, we are interested in efficiently solving the following optimization problem

$$\min_{X_{\partial a \setminus i}} \max \left( \mu_1(x_1), \mu_2(x_2), ..., \mu_K(x_K), f(X_{\partial a \setminus i}, x_i = c_i) \right) \qquad (5)$$

where, without loss of generality we are assuming $\partial a \setminus i = \{1, \ldots, K\}$, and for better readability, we drop the index $a$, in factors $(f_a)$, messages $(\mu_{ka}, \eta_{ai})$ and elsewhere, when it is clear from the context.

There are $2^K$ configurations of $X_{\partial a \setminus i}$, one of which is the minimizing solution. We will divide this set in half in each iteration and save the minimum in one of these halves in the min-max candidate list $\mathcal{C}$. The maximization part of the expression is equivalent to $\max \left( \max \left( \mu_1(x_1), \mu_2(x_2), ..., \mu_K(x_K) \right), f(X_{\partial a}, x_i = c_i) \right)$.

Let $\mu_{j_1}(c_{j_1})$ be the largest $\mu$ value that is obtained at some index $j_1$, for some value $c_{j_1} \in \{0, 1\}$. In other words $\mu_{j_1}(c_{j_1}) = \max \left( \mu_1(0), \mu_1(1), ..., \mu_K(0), \mu_K(1) \right)$. For future use, let $j_2, \ldots, j_K$ be the index of the next largest message indices up to the $K$ largest ones, and let $c_{j_2}, \ldots, c_{j_K}$ be their

corresponding assignment. Note that the same message (*e.g.* $\mu_3(0), \mu_3(1)$) could appear in this sorted list at different locations.

We then partition the set of all assignments to $X_{\partial a \setminus i}$ into two sets of size $2^{K-1}$ depending on the assignment to $x_{j_1}$: 1) $x_{j_1} = c_{j_1}$ or; 2) $x_{j_1} = 1 - c_{j_1}$. The minimization of Eq. (5) can also be divided to two minimizations each having $x_{j_1}$ set to a different value. For $x_{j_1} = c_{j_1}$, Eq. (5) simplifies to

$$\eta^{(j_1)} = \max\left(\mu_{j_1}(c_{j_1}), \min_{X_{\partial a \setminus \{i,j_1\}}} \left(f(X_{\partial a \setminus \{i,j_1\}}, x_i = c_i, x_{j_1} = c_{j_1})\right)\right) \qquad (6)$$

where we need to minimize $f$, subject to a fixed $x_i, x_{j_1}$. We repeat the procedure above at most $K$ times, for $j_1, \ldots, j_k, \ldots j_K$, where at each iteration we obtain a candidate solution, $\eta^{(j_m)}$ that we add to the candidate set $\mathcal{C} = \{\eta^{(j_1)}, \ldots, \eta^{(j_K)}\}$. The final solution is the smallest value in the candidate solution set, $\min \mathcal{C}$.

**Early Termination.** If $j_k = j_{k'}$ for $1 \le k, k' \le K$ it means that we have performed the minimization of Eq. (5) for both $x_{j_k} = 0$ and $x_{j_k} = 1$. This means that we can terminate the iterations and report the minimum in the current candidate set. Adding the cost of sorting $\mathcal{O}(K \log(K))$ to the worst case cost of minimization of $f()$ in Eq. (6) gives a total cost of $\mathcal{O}(K(\log(K) + \omega))$.

**Arbitrary Discrete Variables.** This algorithm is not limited to binary variables. The main difference in dealing with cardinality $D > 2$, is that we run the procedure for at most $K(D-1)$ iterations, and in early termination, all variable values should appear in the top $K(D-1)$ incoming message values.

For some factors, we can go further and calculate *all* factor-to-variable messages leaving $f_a$ in a time linear in $|\partial a|$. The following section derives such update rule for a type of factor that we use in the make-span application of Section 5.2.

### 4.1 Choose-One Constraint

If $f_a(X_{\partial a})$ implements a constraint such that only a subset of configurations $X_{\mathcal{A}} \subset \mathcal{X}_{\partial a}$, of the possible configurations of $X_{\partial a} \in \mathcal{X}_{\partial a}$ are allowed, then the message from function $f_a$ to $x_i$ simplifies to

$$\eta_{ai}(x_i') = \min_{X_{\partial a} \in \mathcal{A}_a | x_i = x_i'} \max_{j \in \partial a \setminus i} \mu_{ja}(x_j) \qquad (7)$$

In many applications, this can be further simplified by taking into account properties of the constraints. Here, we describe such a procedure for factors which enforce that exactly one of their binary variables be set to one and all others to zero. Consider the constraint $f(x_1, ..., x_K) = \delta(\sum_k x_k, 1)$ for binary variables $x_k \in \{0, 1\}$, where $\delta(x, x')$ evaluates to $-\infty$ iff $x = x'$ and $\infty$ otherwise.[2]

Using $X_{\setminus i} = (x_1, x_2, ..., x_{i-1}, x_{i+1}, ..., x_K)$ for $X$ with $x_i$ removed, Eq. (7) becomes

$$\begin{aligned}
\eta_i(x_i) &= \min_{X_{\setminus i} | \sum_{k=1}^{K} x_k = 1} \max_{k | k \ne i} \mu_k(x_k) \\
&= \begin{cases} \max_{k | k \ne i} \mu_k(0) & \text{if} \quad x_i = 1 \\ \min_{X_{\setminus i} \in \{(1,0,...,0),(0,1,...,0),...,(0,0,...,1)\}} \max_{k | k \ne i} \mu_k(x_k) & \text{if} \quad x_i = 0 \end{cases}
\end{aligned} \qquad (8)$$

Naive implementation of the above update is $\mathcal{O}(K^2)$ for each $x_i$, or $\mathcal{O}(K^3)$ for sending messages to all neighbouring $x_i$. However, further simplification is possible. Consider the calculation of $\max_{k | k \ne i} \mu_k(x_k)$ for $X_{\setminus i} = (1, 0, \ldots, 0)$ and $X_{\setminus i} = (0, 1, \ldots, 0)$. All but the first two terms in these two sets are the same (all zero), so most of the comparisons that were made when computing $\max_{k | k \ne i} \mu_k(x_k)$ for the first set, can be reused when computing it for the second set. This extends to all $K - 1$ sets $(1, 0, \ldots, 0), \ldots, (0, 0, \ldots, 1)$, and also extends across the message updates for different $x_i$'s. After examining the shared terms in the maximizations, we see that all that is needed is

$$k_i^{(1)} = \arg \max_{k | k \ne i} \mu_k(0), \quad k_i^{(2)} = \arg \max_{k | k \ne i, k_i^{(1)}} \mu_k(0), \qquad (9)$$

the indices of the maximum and second largest values of $\mu_k(0)$ with $i$ removed from consideration. Note that these can be computed for all neighbouring $x_i$ in time linear in $K$, by finding the top three

values of $\mu_k(0)$ and selecting two of them appropriately depending on whether $\mu_i(0)$ is among the three values. Using this notation, the above update simplifies as follows:

$$\eta_i(x_i) = \begin{cases} \mu_{k_i^{(1)}}(0) & \text{if} \quad x_i = 1 \\ \min\big(\min_{k|k\neq i,k_i^{(1)}} \max(\mu_{k_i^{(1)}}(0), \mu_k(1)), \max(\mu_{k_i^{(1)}}(1), \mu_{k_i^{(2)}}(0))\big) & \text{if} \quad x_j = 0 \end{cases}$$

$$= \begin{cases} \mu_{k_{ai}^{(1)}}(0) & \text{if} \quad x_i = 1 \\ \min\big(\max(\mu_{k_i^{(1)}}(0), \min_{k|k\neq i,k_i^{(1)}} \mu_k(1)), \max(\mu_{k_i^{(1)}}(1), \mu_{k_i^{(2)}}(0))\big) & \text{if} \quad x_i = 0 \end{cases} \tag{10}$$

The term $\min_{k|k\neq i,k_i^{(1)}} \mu_k(1)$ also need not be recomputed for every $x_i$, since terms will be shared. Define the following:

$$s_i = \arg \min_{k\neq i,k_i^{(1)}} \mu_k(1), \tag{11}$$

which is the index of the smallest value of $\mu_k(1)$ with $i$ and $k_i^{(1)}$ removed from consideration. This can be computed efficiently for all $i$ in time that is linear in $K$ by finding the smallest three values of $\mu_k(1)$ and selecting one of them appropriately depending on whether $\mu_i(1)$ and/or $\mu_{k_i^{(1)}}$ are among the three values. The resulting message update for K-choose-1 constraint becomes

$$\eta_i(x_i) = \begin{cases} \mu_{k_i^{(1)}}(0) & \text{if} \quad x_i = 1 \\ \min\big(\max(\mu_{k_i^{(1)}}(0), \mu_{s_i}(1)), \max(\mu_{k_i^{(1)}}(1), \mu_{k_i^{(2)}}(0))\big) & \text{if} \quad x_i = 0 \end{cases} \tag{12}$$

This shows that messages to all neighbouring variables $x_1, ..., x_K$ can be obtained in time that is linear in $K$. This type of constraint also has a tractable form in min-sum and sum-product inference, albeit of a different form (*e.g.* see Gail et al., 1981; Gupta et al., 2007).

## 5 Experiments and Applications

In the first part of this section we compare min-max propagation with the only alternative min-max inference method over FGs that relies on sum-product reduction. In the second part, we formulate the real-world problem of makespan minimization as a min-max inference problem, with high-order factors. In this application, the sum-product reduction is not tractable; to formulate the makespan problem using a FG we need to use high-order factors that do not allow an efficient (polynomial time) sum-product message update. However, min-max propagation can be applied using the efficient updates of the previous section.

### 5.1 Sum-Product Reduction vs. Min-Max Propagation

Like all belief propagation algorithms, min-max propagation is exact when the FG is tree. However, our first point of interest is how min-max propagation performs on loopy graphs. For this, we compare its performance against the sum-product (or CSP) reduction.

Sum-product reduction of (Ravanbakhsh et al., 2014) seeks the min-max value using bisection-search over all values in the range of all factors in the FG – *i.e.* $\mathcal{Y} = \{f_a(X_{\partial a}) \forall a, X_{\partial a}\}$. In each step of the search a value $y \in \mathcal{Y}$ is used to reduce the min-max problem to a CSP. This CSP is satisfiable iff the min-max solution $y^* = \min_X \max_a f_a(X_{\partial a})$ is less than the current $y$. The complexity of this search procedure is $\mathcal{O}(\log(|\mathcal{Y}|)\tau)$, where $\tau$ is the complexity of solving the CSP. Following that paper, we use Perturbed Belief Propagation (PBP) (Ravanbakhsh and Greiner, 2015) to solve the resulting CSPs.

**Experimental Setup.** Our setup is based on the following observations

**Observation 1.** *For any strictly monotonically increasing function $g : \Re \to \Re$,*

$$\arg \min_X \max_a f_a(X_{\partial a}) = \arg \min_X \max_a g(f_a(X_{\partial a}))$$

that is only the *ordering* of the factor values affects the min-max assignment. Using the same argument, application of monotonic $g()$ does not inherently change the behaviour of min-max propagation either.

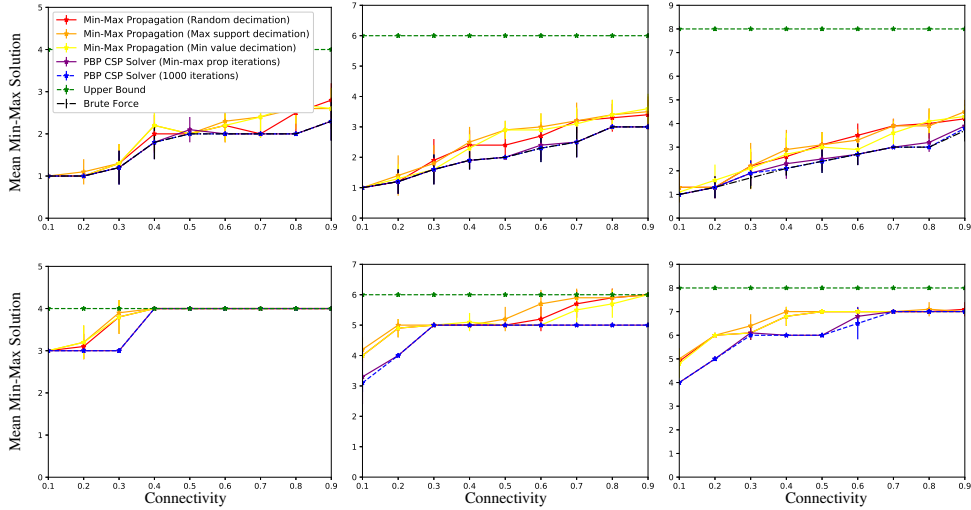

Figure 4: Min-max performance of different methods on Erdos-Renyi random graphs. Top: N=10, Bottom: N=100, Left: D=4, Middle: D=6, Right: D=8.

**Observation 2.** *Only the factor(s) which output(s) the max value,* i.e. ***max factor(s), matter. For all other factors the variables involved can be set in any way as long as the factors' value remains smaller or equal to that of the max factor.*

This means that variables that do not appear in the max factor(s), which we call **free variables**, could potentially assume any value without affecting the min-max value. Free variables can be identified from their uniform min-max marginals. This also means that the min-max assignment is not unique. *This phenomenon is unique to min-max inference and does not appear in min-sum and sum-product counterparts.*

We rely on this observation in designing benchmark random min-max inference problems: i) we use integers as the range of factor values; ii) by selecting all factor values in the same range, we can use the number of factors as a *control parameter* for the difficulty of the inference problem.

For $N$ variables $x_1, \ldots, x_N$, where each $x_i \in \{1, \ldots, D\}$, we draw Erdos-Renyi graphs with edge probability $p \in (0, 1]$ and treat each edge as a pairwise factor. Consider the factor $f_a(x_i, x_j) = \min(\pi(x_i), \pi'(x_j))$, where $\pi, \pi'$ are permutations of $\{1, \ldots, D\}$. With $D = 2$, this definition of factor $f_a$ reduces to 2-SAT factor. This setup for random min-max instances therefore generalizes different K-SAT settings, where the min-max solution of $min_X \max_a f_a(X_{\partial a}) = 1$ for $D = 2$, corresponds to a satisfying assignment. *The same argument with $K > 2$ establishes the "NP-hardness" of min-max inference in factor-graphs.*

We test our setup on graphs with $N \in \{10, 100\}$ variables and cardinality $D \in \{4, 6, 8\}$. For each choice of $D$ and $N$, we run min-max propagation and sum-product reduction for various connectivity in the Erdos-Renyi graph. Both methods use random sequential update. For $N = 10$ we also report the exact min-max solutions.

Min-max propagation is run for a maximum $T = 1000$ iterations or until convergence, whichever comes first. The number of iterations actually taken by min-max propagation are reported in appendix. The PBP used in the sum-product reduction requires a fixed $T$; we report the results for $T$ equal to the worse case min-max convergence iterations (see appendix) and $T = 1000$ iterations. Each setting is repeated 10 times for a random graph of a fixed connectivity value $p \in (0, 1]$.

**Decimation.** To obtain a final min-max assignment we need to fix the free variables. For this we use a decimation scheme similar to what is used with min-sum inference or in finding a satisfying CSP assignment in sum-product. We consider three different decimation procedures:

*Random:* Randomly choose a variable, set it to the state with minimum min-max marginal value.

*Min-value:* Fix the variable with the minimum min-max marginal value.

*Max-support:* Choose the variable for which its min value occurs with the highest frequency.

**Results.** Fig. 4 compares the performance of sum-product reduction that relies on PBP with min-max propagation and brute-force. For min-max propagation we report the results for three different decimation procedures. Each column uses a different variable cardinality $D$. While this changes the range of values in the factors, we observe a similar trend in performance of different methods. In the top row, we also report the exact min-max value. As expected, by increasing the number of factors (connectivity) the min-max value increases. Overall, the sum-product reduction (although asymptotically more expensive), produces slightly better results. Also different decimation schemes do not significantly affect the results in these experiments.

## 5.2 Makespan Minimization

The objective in the makespan problem is to schedule a set of given jobs, each with a load, on machines which operate in parallel such that the total load for the machine which has the largest total load (*i.e.* the makespan) is minimized (Pinedo, 2012). This problem has a range of applications, for example in the energy sector, where the machines represent turbines and the jobs represent electrical power demands.

Given $N$ distinct jobs $\mathcal{N} = \{1, \ldots, n, \ldots, N\}$ and $M$ machines $\mathcal{M} = \{1, \ldots, m, \ldots, M\}$, where $p_{nm}$ represents the load of machine $m$, we denote the assignment variable $x_{nm}$ as whether or not job $n$ is assigned to machine $m$. The task is to find the set of assignments $x_{nm} \, \forall n \in \mathcal{N}, \forall m \in \mathcal{M}$ which minimizes the total cost function below, while satisfying the associated set of constraints:

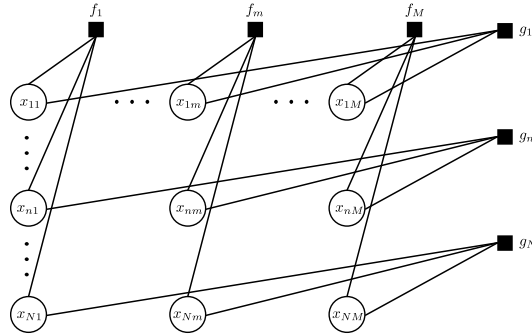

Figure 5: Makespan FG.

$$\min_X \max_m \left( \sum_{n=1}^{N} p_{nm} x_{nm} \right) \quad \text{s.t.} \quad \sum_{m=1}^{M} x_{nm} = 1 \quad x_{nm} \in \{0,1\} \quad \forall n \in \mathcal{N}, m \in \mathcal{M} \quad (13)$$

The makespan minimization problem is NP-hard for $M = 2$ and strongly NP-hard for $M > 2$ (Garey and Johnson, 1979). Two well-known approximation algorithms are the 2-approximation greedy algorithm and the 4/3-approximation greedy algorithm. In the former, all machines are initialized as empty. We then select one job at random and assign it to the machine with least total load given the current job assignments. We re-

Figure 6: Min-max ratio to a lower bound (lower is better) obtained by LPT with 4/3-approximation guarantee versus min-max propagation using different decimation procedures. $N$ is the number of jobs and $M$ is the number of machines. In this setting, all jobs have the same run-time across all machines.

| M | N | LPT | Min-Max Prop (Random Dec.) | Min-Max Prop (Max-Support Dec.) | Min-Max Prop (Min-Value Dec.) |
|---|---|---|---|---|---|
| 8 | 25 | 1.178 | 1.183 | **1.091** | 1.128 |
| | 26 | 1.144 | 1.167 | **1.079** | 1.112 |
| | 33 | 1.135 | 1.144 | **1.081** | 1.093 |
| | 34 | 1.117 | 1.132 | **1.071** | 1.086 |
| | 41 | 1.112 | 1.117 | **1.055** | 1.077 |
| | 42 | 1.094 | 1.109 | 1.079 | **1.074** |
| 10 | 31 | 1.184 | 1.168 | 1.110 | **1.105** |
| | 32 | 1.165 | 1.186 | **1.109** | 1.111 |
| | 41 | 1.138 | 1.183 | **1.077** | 1.088 |
| | 42 | 1.124 | 1.126 | **1.074** | 1.090 |
| | 51 | 1.112 | 1.131 | **1.077** | 1.081 |
| | 52 | 1.102 | 1.100 | **1.051** | 1.076 |

peat this process until no jobs remain. This algorithm is guaranteed to give a schedule with a makespan no more than 2 times larger than the one for the optimal schedule (Behera, 2012; Behera and Laha, 2012) The 4/3-approximation algorithm, a.k.a. the Longest Processing Time (LPT) algorithm, operates similar to the 2-approximation algorithm with the exception that, at each iteration, we always take the job with the next largest load rather than selecting one of the remaining jobs at random (Graham, 1966).

**FG Representation.** Fig. 5 shows the FG with binary variables $x_{nm}$, where the factors are

$$f_m(x_{1m},\ldots,x_{Nm}) = \sum_{n=1}^{N} p_{nm}x_{nm} \,\forall m \quad ; \quad g_n(x_{n1},\ldots,x_{nM}) = \left\{ \begin{array}{ll} 0, & \sum_{m=1}^{M} x_{nm} = 1 \\ \infty, & \text{otherwise} \end{array} \right. \,\forall n$$

where $f()$ computes the total load for a machine according to and $g()$ enforces the constraint in Eq. (13). We see that the following min-max problem over this FG minimizes the makespan

$$\min_X \max \left( \max_m f_m(x_{1m},...,x_{Nm}), \max_n g_n(x_{n1},...,x_{nM}) \right). \tag{14}$$

Using the procedure for passing messages through the $g$ constraints in Section 4.1 and using the procedure of Section 4 for $f$, we can efficiently approximate the min-max solution of Eq. (14) by message passing. Note that the factor $f()$ in the sum-product reduction of this FG has a non-trivial form that does not allow efficient message update.

**Results.** In an initial set of experiments, we compare min-max propagation (with different decimation procedures) with LPT on a set of benchmark experiments designed in (Gupta and Ruiz-Torres, 2001) for the identical machine version of the problem – *i.e.* a task has the same processing time on all machines.

Fig. 6 shows the scenario where min-max prop performs best against the LPT algorithm. We see that this scenario involves large instance (from the additional results in the appendix, we see that our framework does not perform as well on small instances). From this table, we also see that max-support decimation almost always outperforms the other decimation schemes.

Figure 7: Min-max ratio (LP relaxation to that) of min-max propagation versus same for the method of (Vinyals et al., 2013) (higher is better). Mode 0, 1 and 2 corresponds to uncorrelated, machine correlated and machine-task correlated respectively.

| Mode | N/M | (Vinyals et al., 2013) | Min-Max Prop |
|------|-----|------------------------|--------------|
|      | 5   | 0.93(0.03)             | **0.95(0.01)** |
| 0    | 10  | **0.94(0.01)**         | 0.93(0.01)   |
|      | 15  | **0.94(0.00)**         | 0.90(0.01)   |
|      | 5   | **0.90(0.01)**         | 0.86(0.07)   |
| 1    | 10  | **0.90(0.00)**         | 0.88(0.00)   |
|      | 15  | **0.87(0.01)**         | 0.73(0.03)   |
|      | 5   | 0.81(0.01)             | **0.89(0.01)** |
| 2    | 10  | 0.81(0.01)             | **0.89(0.01)** |
|      | 15  | 0.78(0.01)             | **0.86(0.01)** |

We then test the min-max propagation with max-support decimation against a more difficult version of the problem: the unrelated machine model, where each job has a different processing time on each machine. Specifically, we compare our method against that of (Vinyals et al., 2013) which also uses distributive law for min-max inference to solve a load balancing problem. However, that paper studies a sparsified version of the unrelated machines problem where tasks are restricted to a subset of machines (*i.e.* they have infinite processing time for particular machines). This restriction, allows for decomposition of their loopy graph into an almost equivalent tree structure, something which cannot be done in the general setting. Nevertheless, we can still compare their results to what we can achieve using min-max propagation with infinite-time constraints.

We use the same problem setup with three different ways of generating the processing times (uncorrelated, machine correlated, and machine/task correlated) and compare our answers to IBM's CPLEX solver exactly as the authors do in that paper (where a high ratio is better). Fig. 7 shows a subset of results. Here again, min-max propagation works best for large instances. Overall, despite the generality of our approach the results are comparable.

## 6 Conclusion

This paper demonstrates that FGs are well suited to model min-max optimization problems with factorization characteristics. To solve such problems we introduced and evaluated min-max propagation, a variation of the well-known belief propagation algorithm. In particular, we introduced an efficient procedure for passing min-max messages through high-order factors that applies to a wide range of functions. This procedure equips min-max propagation with an ammunition unavailable to min-sum and sum-product message passing and it could enable its application to a wide range of problems. In this work we demonstrated how to leverage efficient min-max-propagation at the presence of high-order factors, in approximating the NP-hard problem of makespan. In the future, we plan to investigate the application of min-max propagation to a variety of combinatorial problems, known as bottleneck problems (Edmonds and Fulkerson, 1970) that can be naturally formulated as min-max inference problems over FGs.

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

## Footnotes

[1] Here we show that solving the minimization problem on any particular factor can be solved in a fixed amount of time. In many applications, doing this might itself involve running another entire inference algorithm. However, please note that our algorithm is agnostic to such choices for optimization of individual factors.

[2]Similar to any other semiring, $\pm\infty$ as the identities of min and max have a special role in defining constraints.
