[Supplementary Material]

# Supplementary Material: Min-Max Propagation

**Christopher Srinivasa**
University of Toronto
Borealis AI
christopher.srinivasa
@gmail.com

**Inmar Givoni**
University of
Toronto
inmar.givoni
@gmail.com

**Siamak Ravanbakhsh**
University of
British
Columbia
siamakx@cs.ubc.ca

**Brendan J. Frey**
University of Toronto
Vector Institute
Deep Genomics
frey@psi.toronto.edu

## Abstract

This is the supplementary material for the paper titled "Min-Max Propagation".

## 1   Functions $f$ that are Easy to Minimize

Here we address the issues of minimizing $f(x)$ subject to a given set of constraints, fixing values of certain $x_i$s. There are several forms of $f$ for which this evaluation is easy. We explain some examples.

**Functions of the form** $f(x) = \sum_i \alpha_i x_i$**.** In the simplest case, where $\alpha_i = 1$, we are counting the number of variables set to one. Therefore, the minimum is obtained by setting all unconstrained variables to zero. The value obtained is equal to the number of constrained variables already fixed to one. With positive coefficients, again, the minimum is obtained by setting all the unconstrained variables to zero, and the value obtained is the weighted sum of the constrained variables fixed to one. Thus, generally, unconstrained variables are fixed to zero if their coefficient is positive, and to one if it is negative. The value obtained is the weighted sum of all coefficients whose variables are fixed to one, constrained and unconstrained.

**Functions of the form** $f(x) = g\left(\sum_i \alpha_i x_i\right)$**.** These can be solved if $g$ is either monotonically increasing or monotonically decreasing. In the former case, all unconstrained variables with positive coefficients are set to zero. All unconstrained variables with negative coefficients are set to one. The value is found by calculating the corresponding $g$ value. In the latter case, all unconstrained variables with positive coefficients are set to one, and all unconstrained variables with negative coefficients are set to zero.

## 2   Additional Results

### 2.1   Sum-Product Reduction vs. Min-Max Propagation

Fig. 1 shows the number of iterations for each of the head-to-head min-max results reported in the paper.

### 2.2   Additional Results for Identical Machines Problems

We compare min-max propagation (with different decimation procedures) against LPT on a set of benchmark experiments designed in (Gupta and Ruiz-Torres, 2001) for the identical machine version of the problem – *i.e.* a task has the same processing time on all machines ($p_n = p_{1n} = p_{2n} = .. = p_{Mn}$).

In total there are four families our benchmark experiments, as shown in Table 1, where for each family all combinations of specified number of jobs $N$, machines $M$, and distributions from which

Figure 1: Performance of different methods on Erdos-Renyi graphs.Top: N=10, Bottom: N=100, Left: K=4, Middle: K=6, Right: K=8.

|     | M         | N                | Dist                            |
|-----|-----------|------------------|---------------------------------|
| E1  | 3,4,5     | 2M,3M,5M         | U(1,20), U(20,50)               |
| E2  | 2,3,4,6,8,10 | 10,30,50,100  | U(100,800)                      |
| E3  | 3,5,8,10  | 3M+1,3M+2,4M+1,  | U(1,100),U(100,200)             |
|     |           | 4M+2, and 5M+2   |                                 |
| E4  | 2         | 9                | U(1,20), U(20,50), U(50,100),   |
|     | 3         | 10               | U(100,200), U(100,800)          |

Table 1: Set of benchmark experiments for identical machines makespan minimization.

the jobs are drawn are tested. The only exception to this is $E4$ where the $M = 2, N = 9$ combination is tested separately from the $M = 3, N = 10$. For all experiments the processing times are drawn from a uniform distribution with varying bounds.

The experiments are designed to set different difficulties of the problem. For example, $E1$ and $E4$ are designed to test small problem instances. In contrast, $E2$ and $E3$ are setup such as to test larger instance. For all experiments, the performance number reported is the ratio of the makespan achieved by the method being tested to a lower bound $LB$ computed as:

$$LB = \max \left( \max_{1 \leq i \leq n} p_n, \sum_{n=1}^{N} p_n/M \right). \tag{1}$$

Table 2 shows the scenario where min-max propagation performs best against the LPT algorithm. We see that this scenario involves large instances. From this table, we also see that max-support decimation almost always outperforms the other decimation schemes. From the additional results in Tables 3 to 7, we see that our framework does not perform as well on small instances.

## 2.3 Additional Results for Unrelated Machines Problem

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

Table 6: Makespan minimization experiment 3 with U(1,100).

| M,N | U(,) | LPT | Min-Max Prop (Random Dec.) | Min-Max Prop (Max-Support Dec.) | Min-Max Prop (Min-Value Dec.) |
|---|---|---|---|---|---|
| 2,9 | U(1,20) | **1.016** | 1.036 | 1.026 | 1.026 |
| | U(20,50) | 1.062 | 1.048 | **1.028** | **1.028** |
| | U(1,100) | 1.017 | **1.016** | 1.034 | 1.034 |
| | U(50,100) | 1.072 | 1.055 | **1.042** | **1.042** |
| | U(100,200) | 1.073 | **1.051** | 1.055 | 1.055 |
| | U(100,800) | 1.028 | 1.027 | **1.026** | **1.026** |
| 3,10 | U(1,20) | **1.031** | 1.127 | 1.057 | 1.076 |
| | U(20,50) | 1.103 | 1.095 | **1.052** | 1.075 |
| | U(1,100) | **1.033** | 1.096 | 1.039 | 1.048 |
| | U(50,100) | 1.142 | 1.120 | **1.083** | 1.068 |
| | U(100,200) | 1.127 | 1.133 | **1.083** | 1.094 |
| | U(100,800) | **1.049** | 1.056 | 1.061 | 1.078 |

Table 7: Makespan minimization experiment 4.

| Mode | Dist |
|---|---|
| Uncorrelated (Mode 0) | $N(100, 10)$ |
| Machine correlated (Mode 1) | $N(\alpha_i, 10)$ where $\alpha_i \sim U(50, 100)$ and $\alpha_i$ is different for each machine |
| Machine/Task correlated (Mode 2) | $N(\beta_k + \alpha_i, 10)$ where $\alpha_i, \beta_k \sim U(50, 100)$, $\alpha_i$ is different for each machine, and $\beta_i$ is different for each job |

Table 8: Processing time generation for unrelated machine makespan minimization.

| Mode | N/M | (Vinyals et al., 2013) | Min-Max Prop |
|---|---|---|---|
| 0 | 10 | **0.94(0.01)** | 0.92(0.01) |
| | 15 | **0.93(0.00)** | 0.90(0.02) |
| 1 | 10 | **0.92(0.00)** | 0.87(0.05) |
| | 15 | **0.89(0.00)** | 0.82(0.02) |
| 2 | 10 | **0.87(0.00)** | 0.85(0.05) |
| | 15 | **0.85(0.00)** | **0.85(0.04)** |

Table 9: Makespan minimization with M=40. Min-max ratio (LP relaxation to that) of min-max propagation versus same for the method of (Vinyals et al., 2013) (higher is better). Mode 0, 1 and 2 corresponds to uncorrelated, machine correlated and machine-task correlated respectively.

still compare their results to what we can achieve using min-max propagation using infinite-time constraints.

We use the same problem setup with three different ways of generating the processing times (uncorrelated, machine correlated, and machine/task correlated) and compare our answers to IBM's CPLEX solver exactly as the authors do in that paper (where a high ratio is better). Table 8 shows a summary of the processing time generation for each of the different ways (note that all normal distributions are bounded at the low end to have a minimum of 10).

Tables 9 to 11 show the results. Here again, min-max propagation works best for large instances.

| Mode | N/M | (Vinyals et al., 2013) | Min-Max Prop |
|---|---|---|---|
| 0 | 10 | **0.94(0.01)** | 0.93(0.00) |
| | 15 | **0.94(0.00)** | 0.89(0.03) |
| 1 | 10 | **0.90(0.00)** | 0.87(0.01) |
| | 15 | **0.87(0.01)** | 0.85(0.02) |
| 2 | 10 | 0.87(0.01) | **0.89(0.03)** |
| | 15 | **0.87(0.01)** | 0.79(0.09) |

Table 10: Makespan minimization with M=60.

| Mode | N/M | (Vinyals et al., 2013) | Min-Max Prop |
|------|-----|------------------------|--------------|
| 0    | 5   | 0.93(0.03)             | **0.95(0.01)** |
|      | 10  | **0.94(0.01)**         | 0.93(0.01)   |
|      | 15  | **0.94(0.00)**         | 0.90(0.01)   |
| 1    | 5   | **0.90(0.01)**         | 0.86(0.07)   |
|      | 10  | **0.90(0.00)**         | 0.88(0.00)   |
|      | 15  | **0.87(0.01)**         | 0.73(0.03)   |
| 2    | 5   | 0.81(0.01)             | **0.89(0.01)** |
|      | 10  | 0.81(0.01)             | **0.89(0.01)** |
|      | 15  | 0.78(0.01)             | **0.86(0.01)** |

Table 11: Makespan minimization with M=80.