[Reviews · NeurIPS 2017]

Reviewer 1



The authors describe the implementation and application of belief propagation for the min,max semiring. While BP is known to work in any commutative semiring, the authors derive efficient updates over the min-max semiring and provide an example application for which the algorithm is relevant. The paper is well-written and easy to follow. I'm a bit on the fence with this paper. The algorithm isn't super novel, though the efficient implementation is, as far as I know, new. I suppose that I would be more likely to vote for acceptance if the authors had compared against more standard BP implementations and explained the trade-offs. For example, min_x max_a f_a(x_a) can be reformulated only as a minimization problem by introducing a new variable y and solving min_{x,y} y such that y >= f_a(x_a). I feel like the message passing updates can also be simplified in this case using the same kinds of ideas that the authors propose and it doesn't even require solving a min max for each update.

Reviewer 2



POST-REBUTTAL UPDATE ==================== I have read and considered the authors' response and stand by my initial decision. I would heavily stress that the revised paper more clearly state prior work on message passing specific to the min-max semiring, including Aji and McEliece and Vinyals et al. SUMMARY: ======== The authors propose a dynamic programming algorithm for solving min-max problems with objectives that decompose as a set of lower dimensional factors. For cases where factors have high arity the authors further show an efficient updating scheme in cases where factors can be efficiently minimized with some variables clamped. Further speedups are shown for cases where the domain is constrained to a feasible subset. PROS: ===== This reviewer found the paper to be well-rounded with a clear presentation of algorithmic components and with experiments on an interesting load balancing application. Results on the latter applications, while not a decisive improvement, are comparable to algorithms specifically taylored to that problem. The method of efficient updating for large clique factors is a good practical consideration. CONS: ===== The strongest criticism of this reviewer is that the authors could more honestly credit Vinyals et al. (2013) with the development of the min-max BP variant. This reviewer is aware that the aforementioned work is not heavily cited and that the authors do cite it as well as compare to experimental results. However, from reading this paper it was not clear that much of the BP formulation was provided by Vinyals et al. Nevertheless, this reviewer feels that the present work provides a number of improvements over Vinyals et al. that merit publication, including a clearer description of min-max BP for factor graphs without specialization to junction trees and efficient updating for high arity factors. Detailed comments: * (L:18) cite Kschischang et al. (2001) in reference to turbo codes for LDPC * A brief statement of min-max semiring existence would be useful (c.f. Vinyals et al. 2013). Sum/max-product and max/min-sum semirings are well-known in the graphical models literature but min-max is not commonly discussed * (L:183-186) Seems to suggest that argmin X is not unique as some elements do not affect objective value at the minimum. Please clarify or discuss to what extent this is a problem * In Sec. 5.2 numerous locations interchange maximization subscript "a" with "m", particularly in Eq. (13) * Fig. 5: Legend is much too small to read on printed version

Reviewer 3



As the title suggests, this paper is interested in solving min-max problems using message-passing algorithms. The starting point is Eq. 1: You want to optimize a vector x such that the max over a discrete set of functions is minimize. The set of functions should be finite. It's well known that changing the semiring can change the message-passing algorithm from a joint optimization (max product) algorithm to a marginalization algorithm (sum product). So, just like the (+,*) semiring yields the sum-product algorithm and the (max,*) semiring yields the max-product algorithm, this paper uses the (min,max) semi-ring. Then, Eqs. 2-3 are exactly the standard BP messages on factors and single variables, just with the new semi-ring. (It would be good if the paper would emphasize more strongly that Eqs. 2 and 3 are exactly analogous-- I had to work out the details myself to verify this.) Since that's a semi-ring you get all the normal properties (exactness on trees, running times). However, you don't seem to inherit much of the folk theorems around using BP on loopy graphs when you change the semi-ring (where most applications probably lie) so this paper investigates that experimentally. The paper assumes that solving the minimization problem on any particular factor can be solved in a fixed amount of time. This is reasonable to make analysis tractable, but it's worth pointing out that in many applications, doing this might itself involve running another entire inference algorithm (e.g. max-product BP!) There is also an algorithmic specialization to speed up message-computation for the case where there is a special "choose-one" form for the factors, similar to some factor forms that are efficient with different semigroups. Experimentally, it is compared to a previous algorithm that runs a bisection search with a CSP solver in each iteration. This appears to perform slightly better, although apparently at higher cost. As far as I know, this particular semi-ring variant of belief propagation has not been investigated before. I don't see strong evidence of a huge practical contribution in the expeirments (though that might reflect my lack of expertise on these problems). Nevertheless, the algorithm is extremely simple in concept, and it certainly seems worth noting that there is another semi-ring for BP that solves another possibly interesting set of problems. Even if the results here aren't that strong in the current form, this might be a first step towards using some of the many improvements that have been made to other semi-ring versions of BP. (I was surprised that this particular semi-ring hasn't been proposed before, but I'm not aware of such a use, and a brief search did not find anything. If this is not actually novel, I would downgrade my rating of this paper.) - Fig. 5 is very hard to read-- illegible in print - In eq. 13 I believe you want a -> m? EDIT AFTER REBUTTAL: I've read the rebuttal and other reviews, see no reason to change my recommendation. (I remain positive.)